# Does Training Motivation Influence Resilience Training Outcome on Chronic Stress? Results from an Interventional Study

**DOI:** 10.3390/ijerph19106179

**Published:** 2022-05-19

**Authors:** Madlaina Niederhauser, Regula Zueger, Sandra Sefidan, Hubert Annen, Serge Brand, Dena Sadeghi-Bahmani

**Affiliations:** 1Military Academy, Swiss Federal Institute of Technology ETH Zurich, 8903 Birmensdorf, Switzerland; madlaina.niederhauser@milak.ethz.ch (M.N.); regula.zueger@milak.ethz.ch (R.Z.); s.sefidan@hotmail.com (S.S.); 2Sleep Disorders Research Center, Kermanshah University of Medical Sciences, Kermanshah 67146, Iran; serge.brand@upk.ch (S.B.); bahmanid@stanford.edu (D.S.-B.); 3Substance Abuse Prevention Research Center, Health Institute, Kermanshah University of Medical Sciences, Kermanshah 67146, Iran; 4School of Medicine, Tehran University of Medical Sciences, Tehran 25529, Iran; 5Division of Sport Science and Psychosocial Health, Department of Sport, Exercise and Health, University of Basel, 4052 Basel, Switzerland; 6Department of Psychology, Stanford University, Stanford, CA 94305, USA; 7Center for Affective, Stress and Sleep Disorders (ZASS), Psychiatric University Hospital Basel, 4002 Basel, Switzerland

**Keywords:** resilience training, training motivation, chronic stress, symptoms of depression, vital exhaustion

## Abstract

Resilience is understood as an acquired skill which aids in coping with acute and chronic stress. Accordingly, the present study aimed to determine the effect of resilience training on mental health problems during chronic stress. To this end, we conducted a quasi-experimental study with 127 male cadets (mean age: 21 years) of the Swiss Armed Forces officers’ school. Whereas the intervention group (IG) received resilience training in addition to the standard officer’s education program, the control group (CG) completed the officers’ school as usual. Data assessment included pre- and post- measurement of chronic stress, symptoms of depression, and vital exhaustion in both groups. Motivation for training was collected before the first training session. Those who received the resilience training reported no change in chronic stress, whereas participants in the CG showed a significant increase in chronic stress over time (*η_p_*^2^ = 0.025). Furthermore, significant differences between IG and CG were only found for symptoms of depression: Participants in the IG reported significantly decreased symptoms of depression, while this was not the case for participants in the CG. Within the IG, participants’ training motivation strongly influenced the effectiveness of the resilience training. More specifically, motivated individuals were more likely to benefit from the resilience training than unmotivated ones. Outcome data suggest that resilience training appeared to favorably affect chronic stress and related mental health symptoms; however, the motivation for the training seemed to be an essential prerequisite.

## 1. Introduction

Stress is defined as a subjectively perceived imbalance between demands and the range of possibilities to respond adequately to such demands; accordingly, stress occurs when personal psychological and/or environmental demands appear difficult to cope with [1]. Stress influences numerous physiological and psychological processes [2]; especially chronic stress, which can lead to a variety of unfavorable psychological consequences and has been associated with an increased risk of developing mental disorders [3,4]. From the view point of psychopathology, and to name just three, posttraumatic stress disorder, major depressive disorder and anxiety disorder may be considered “stress-related disorders” [5]. Chronic stress exposure can further result in vital exhaustion, characterized by a triad of increased irritability, feelings of demoralization and excessive fatigue and energy loss [6]. Given this background, psychological research explored the possibilities to improve important skills to deal with stress and to become more resilient [7,8,9].

One line of research about stress focused on resilience. Resilience describes how people maintain their mental health despite exposure to psychological adversity [10,11,12]. Resilience equals the ability to adapt positively to stressful circumstances [13,14] and to remain functionally stable and well despite ongoing stress [15,16]. Resilience consists of numerous behavioral, cognitive, affective and social factors which protect individuals in the face of adversity, and resilience has a positive effect on mental health [17]. Resilience is recognized as an acquired skill which can be developed and improved upon; given this, resilience training programs were designed [8,18,19]. Resilience training programs are created to teach individual skills to promote mental health, to enhance adaptation to stress and to prevent adjustment problems [20]. The effect of such programs was evaluated with different populations, formats, durations, and settings, and a wide range of outcomes have been examined [18,19]. Meta-analyses examining the effectiveness of resilience training demonstrated that the overall effect of such programs was small to moderate [8,19]. One-on-one or classroom-based programs were more effective compared with the train-the-trainer and computer-based delivery formats [19].

To accurately assess resilience and the effect of resilience training, confronting participants with a significant challenge or stressor during the study period appeared to be the best setting. Resilience and the effect of resilience training could be particularly accurately determined if an individual was exposed to stress [18] and if they were challenged to maintain their mental or physical health in such situations [11]. However, most of the abovementioned studies lacked stressors during the intervention and study periods [18,19,21]. Only a few studies focused on the effect of resilience training in acute stress situations such as police interventions [22] or standardized acute stress tests (Trier Social Stress Test (TSST) [23,24]. Likewise, interventions focusing on chronic stress appeared to be methodologically more demanding. So far, only a few studies have investigated the influence of resilience training in such chronic stress settings, and all of them were conducted with police officers (e.g., [25]) or within military organizations (e.g., [26,27]).

With regard to military settings, military training is a multi-stressor environment, and military populations are generally at greater risk of experiencing substantial stress and adversity [28,29,30]. The stressors in question include sleep deprivation [31], environmental challenges, psychological strain [32,33], a highly structured environment, the lack of personal autonomy [32,34], and dramatic changes in living arrangements [32,35]. Furthermore, earlier sources of support such as friends and families are no longer directly available [32]. Furthermore, all recruits or cadets attend the same school completing the same tasks and living in the same environment. This supports the view that a military training has elements that support a more objective assessment [27] and therefore provides a controlled and excellent framework to examine factors associated with stress, mental health, and resilience.

The effectiveness of the resilience training in military contexts has been shown for different outcomes, but with mixed results. Williams et al. (2007) found that training participants exhibited higher scores in group cohesion, problem-solving coping strategies, and perceived social support after a military training. Furthermore, resilience training led to lower scores for anger expression, coping strategies, conflicts in relationships, perceived stress during the last week, and emotional stress response [36]. One study showed the positive effects of a resilience training participation on burnout, organizational stress, alcohol use, aggression, and psychological flexibility after the training [25]. Cohn and Pakenham (2008) found less reliance on self-blame coping and a better adjustment to the stressful situation in the intervention group compared to the control condition. However, no effects between the two groups were found for attribution styles, the expectancy of control, and other coping styles after the resilience training [37].

Resilience training with a focus on cognitive skills revealed positive effects on cognitive skills, performance, and physical fitness [20]. Furthermore, self-reflection training helped to decrease depression and anxiety symptoms and perceived stressor frequency [27] compared to control groups. Also, social cognition (empathy, perspective taking and military hardiness) improved [38]. In contrast, one study found no effect at all on mental health, well-being, attitudes to mental illness, help-seeking, cohesion, or on perceptions of leadership or military variables [39], and results of a meta-analysis on the influence of resilience training on different outcomes showed only small effect sizes [19].

The mixed results and small effect sizes of previous studies suggested that there may be further and latent factors influencing the success of resilience training. Indeed, external factors such as training settings and delivery format influenced the resilience training effectiveness [19]. And as mentioned, one-on-one or classroom-based formats yielded the highest effects. On the other hand, we also note that no previous research investigated the impact of individual factors like motivation, self-reflection, or personality on training outcomes.

In regard to motivation, it plays a central role for changing a dysfunctional behavior and for replacing it with health-promoting behavior [40]. Indeed, the trainees’ motivation was one of the critical determinants in learning and the transfer of learned issues [41]. Motivation describes the desire to learn the training content; motivation is based on the conviction that efforts during the training will influence learning and that such efforts will be helpful to achieve the desired results [42]. Studies among private companies with internal training showed that individual differences among trainees in motivation influenced both training transfer [43] and training effectiveness [44].

Furthermore, the importance of motivation is also shown in the therapeutic setting, where motivation is an essential predictor for behavioral change and positive outcomes [45,46]. Thus, it appeared that the successful implementation of resilience training could depend on trainees’ motivation; however, no study has examined the influence of trainees’ motivation on the effectiveness of resilience training so far.

To summarize, a small number of studies have examined the effects of a resilience training in a chronic stress setting, especially in a standardized environment. Until now, to our knowledge, no study has investigated the effects of resilience training on chronic stress and vital exhaustion. Furthermore, the influence of training motivation on the effectiveness of a resilience training in military settings has not yet been studied. Therefore, the purpose of the present study was to extend the current psychological resilience research; to this end, we investigated the influence of resilience training on chronic stress, symptoms of depression and vital exhaustion in a standardized chronic stress setting, and we tested the influence of training motivation. First, we hypothesized that resilience training participation will decrease chronic stress, symptoms of depression and vital exhaustion compared to a control group. Second, we expected a positive influence of motivation for resilience training on its outcomes.

## 2. Materials and Methods

### 2.1. Participants and Study Design

The current study was part of a longitudinal research project during officers’ school in the Swiss Armed Forces. Switzerland has compulsory military service for men, with a complex recruitment process consisting of physical examinations and psychological screening for basic military training. During the recruitment process, the best recruits are enlisted for voluntary continuation as cadre and are first trained in the non-commissioned officers’ school and later in the officers’ school (OS). The OS lasts fifteen weeks and is extraordinarily intense. The training repeatedly pushes the cadets to their physical and mental limits and can be considered to be a highly stressful environment.

The study consisted of 161 cadets starting in two different officers’ schools between 2016 and 2017. After a brief introduction to the study, volunteering cadets signed the written consent form. The cadets voluntarily agreed and were not paid for the participation. To avoid cross-contamination of training content (Cacioppo et al., 2015), each OS was entirely conducted as an intervention (IG) or control group (CG). The intervention group received resilience training during the OS (*n_IG_* = 73), while the control group completed the school conventionally without any resilience training (*n_CG_* = 88).

The Ethics Committee Zurich (“Kantonale Ethikkommission KEK”; Req-2016-00465) reviewed the study, which was conducted according to the current and seventh revision of the Ethical Principles of the Declaration of Helsinki for experiments involving human subjects (World Medical Association, 2013).

### 2.2. Intervention

The resilience training was carried out four times as 90-min sessions, once a week between the fourth and seventh week of the OS. During the rest of the week, characterized by demanding military, physical fitness, and leadership training, the participants practiced the learned skills and techniques and completed their homework. The resilience training consisted of short theoretical lectures and exercises in small groups led by moderators. The group composition and moderator remained unaltered throughout the intervention. The moderators were trained to support participants, moderate discussions, and encourage self-reflection.

Each session of the Army resilience training set different thematic priorities. In the first week, participants learned to analyze their behavior and their individual contributions in stressful situations using a new conceptualized model based on a combination of previous models [47,48]. Furthermore, participants learned to uncover selected thinking traps [49] and a new conceptualized optimism exercise. Here, the cadets were instructed to make a knot in a string each time they experienced something positive and then to take the string out from time to time to remind themselves of the positive experiences in life. In the second session, the participants identified their values and core beliefs. They learned to recognize individual triggers [49] and deal with negative thoughts [50]. The third session thematized individual coping strategies [1,51] and the cadets learned how to achieve optimal performance through energy management [50,52,53,54]. The last session covered communication styles and character strengths [55,56,57] and a “bombardment of strengths” as a final exercise [58].

### 2.3. Measures

Pre-measurements were taken at the end of the first week of OS, two days before the training started. Post-measurements were made at the end of the seventh week, five days after the last training session.

#### 2.3.1. Chronic Stress

Chronic stress was measured with the perceived stress questionnaire (PSQ [59]). Twenty-five items assess subjective experiences of stressful situations in the previous month and stress reactions on cognitive and emotional levels (e.g., “I have too many things to do”). Items are rated on a 4-point Likert-scale ranging from 1 (*rarely*) to 4 (*usually*). Higher scores reflect a more pronounced chronic stress. The internal consistency was satisfactory (α = 0.84).

#### 2.3.2. Vital Exhaustion

Vital exhaustion was assessed with the short version of the Maastricht VE Questionnaire (MQ [60]). Each item (e.g., “Do you often feel tired?”) was scored between *no* (=0), *uncertain* (=1), and *yes* (=2). Higher sum scores reflect a more pronounced vital exhaustion. The current sample had an acceptable consistency (Cronbach’s α = 0.75).

#### 2.3.3. Symptoms of Depression

Symptoms of depression were measured by the general depression scale (German version; ADS [61]). Twenty items (e.g., “During the last week I could not get rid of my gloomy mood”) reflect symptoms of depression and were summed up to a total depression score. Each item is rated on a four-point Likert-Scale ranging from 0 (*rarely or not at all*) to 3 (*usually, all the time*), with higher sum/mean scores reflecting higher symptoms of depression. The current sample had an acceptable internal consistency (Cronbach’s α = 0.75).

#### 2.3.4. Training Motivation

Training motivation was measured with a two-item questionnaire (designed for research) immediately before the training started. One item represented the general motivation (i.e., “How motivated are you to participate in this resilience training?”) and one the learning motivation (“How motivated are you to profit from this resilience training”). Both items were measured on a visual analog scale (VAS [62] which allows a good assessment of the whole motivation continuum. The items ranged from 0 (*not at all*) to 100 (*very*), with higher scores reflecting a higher motivation. The two items correlated very high (*r* = 0.75, *p* < 0.001) and had a good consistency (Cronbach’s α = 0.85).

### 2.4. Statistical Analysis

First, we calculated whether dropouts and remainders systematically differ; to this end, we performed a series of Welch’s *t*-tests and Chi-square tests.

Second, a series of Pearson’s correlations was performed to identify associations between motivation and chronic stress, vital exhaustion and symptoms of depression.

Third, to predict the influence of motivation on changes in chronic stress, vital exhaustion and symptoms of depression over time (measured with delta values), three regression analyses were performed.

Fourth, a series of ANOVAs for repeated measure was conducted with the following factors: time (pre- and posttest), group (intervention vs. control group) and the time × group interactions. The homogeneity of error variances was tested with Levene’s test, and homogeneity of covariance was tested with a box test using a significance level *p* < 0.001, as suggested by different authors [63,64]. All preliminary conditions were met.

Fifth, to test the influence of training motivation on chronic stress, vital exhaustion and symptoms of depression, we performed regression analysis and ANOVAs for repeated measures. For the latter, the intervention group was divided with median split in high and low motivated participants and conducted with the following factors: time (pre- and post-test), group (motivation high, motivation low, control group) and the time × group interactions.

Sixth, to identify participants with significant individual changes over time in chronic stress, vital exhaustion and symptoms of depression, the reliable change index was calculated (RCI [65]). The RCI is a value that reflects the individual change over time and determined significant changes from pre- to posttest. Change values of 0.50 for chronic stress, 8 for vital exhaustion and 9 for symptoms of depression were calculated in the present study as significant individual change. If there were significant differences between the groups in RCI, Chi-square tests were conducted.

Effect sizes for the F-statistics were reported as partial eta-squared [*η_p_*^2^]) and interpreted as follows: trivial (T) 0.019 < *η_p_*^2^; small (S) = 0.020 ≤ *η_p_*^2^ ≤ 0.059, medium (M) = 0.06 ≤ *η_p_*^2^ ≤ 0.139, or large (L) = *η_p_*^2^ ≥ 0.14 [66,67].

The level of significance for all tests was set at *p* < 0.05. All analyses were performed with SPSS^®^ 27.0 (IBM Corporation, Armonk, NY, USA) for Windows.

## 3. Results

### 3.1. Participants’ Characteristics

Participants were 161 cadets of the OS. Four female participants were excluded to simplify interpretation and because the number of women were not balanced between groups. Eight participants were excluded because of missing data and seven because of unserious answers (measured with the ADS lie subscale). Some participants quit officers’ school (*n* = 15) because of medical or military reasons. Those quitting officers’ school (*M* = 0.35, *SD* = 0.14) had slightly higher scores in chronic stress (*t*(16.67) = −1.97, *p* = 0.07, *d* = 0.53) than those remaining (*M* = 0.28, *SD* = 0.12). Furthermore, there were marginally significant differences in vital exhaustion at pretest (*t*(15.72) = −1.77, *p* = 0.09, *d* = 0.54) with lower scores for remainders (*M* = 4.29, *SD* = 3.59) versus the dropouts (*M* = 6.67, *SD* = 5.04), but not significant differences in symptoms of depression (*t* < 1.00). These results were not surprising, as the OS was a very stressful experience and less resilient people were not up to the requirements. Furthermore, there were no significant differences in the number of dropouts between the CG and the IG, and thus the comparative analysis could be continued.

The mean age of the 127 remaining participants was 21 years (*M* = 20.94, *SD* = 1.58), ranging from 18 to 29 years. Table 1 provides the descriptive statistical overview for the intervention and control group.

### 3.2. Differences between the Intervention and Control Group in Chronic Stress, Vital Exhaustion and Symptoms of Depression

Table 2 provides the descriptive and inferential statistical indices for chronic stress, vital exhaustion and symptoms of depression separately for intervention and control group at pre- and post-test. The differences between the intervention and the control group in pre- and post-tests were measured for chronic stress, vital exhaustion and symptoms of depression. The intervention group has higher values in all variables at pretest. The only significant interaction term of time by group was found for symptoms of depression; symptoms of depression of the intervention group decreased over time compared to the control group, but the effect size was small.

### 3.3. Correlational Analysis between Motivation, Chronic Stress, Vital Exhaustion and Symptoms of Depression

Table 3 provides an overview of correlation coefficients between motivation before the training and chronic stress, vital exhaustion, and symptoms of depression, both at pre- and post-test. All correlation with training motivation were made only with intervention group participants. Training motivation did not correlate with any of the dependent variables (chronic stress, vital exhaustion, symptoms of depression) at pre- or post-test (Table 3). Changes in chronic stress were correlated with changes in vital exhaustion (r = 0.290, *p* = 0.001) as well as depression symptoms (r = 0.216, *p* = 0.015).

### 3.4. Regression Analysis with Motivation as a Predictor for Changes in Chronic Stress, Vital Exhaustion, and Symptoms of Depression

As a next step, a series of regression analyses was performed to test the influence of training motivation on changes (delta values from pre- to post) in chronic stress, vital exhaustion and symptoms of depression. Motivation (*β* = −0.293, *p* = 0.02) was a significant predictor for the relationship between pre- and post-stress (*F*(2, 61) = 5.631, *p* = 0.021, *R*^2^ = 0.086). Thus, the higher the motivation of participants was, the lower was the increase in chronic stress from pre- to post-test. For changes in vital exhaustion between pre- and posttest, training motivation (*β* = −0.361, *p* = 0.01) was an important predictor (*F*(2, 61) = 9.00, *p* = 0.01, *R*^2^ = 0.13), shown by the fact that higher motivation was related to a lower increase in vital exhaustion from pre- to posttest and, finally, motivation (*β* = −0.259, *p* = 0.04) was a significant predictor for changes in symptoms of depression (*F*(2, 61) = 4.30, *p* = 0.04, *R*^2^ = 0.067), so that the higher the motivation was, the lower was the increase in symptoms of depression from pre- to post-test.

### 3.5. Influence of Motivation on Resilience Training Effects

To test the influence of motivation in more detail and to find out more about the influence of motivation on resilience training effects, the intervention group was divided with the median split into high and low motivated participants.

Participants of the control group and those reporting lower scores in motivation had an increase in chronic stress, but no changes in vital exhaustion and symptoms of depression over time. In contrast, highly motivated participants had no change in chronic stress, but a decrease of vital exhaustion and symptoms of depression. Table 4 provides the results for all repeated measure ANOVAs and Figure 1, Figure 2 and Figure 3 illustrate these results.

### 3.6. Reliable Change Index

The frequency of individuals who demonstrated significant change was compared between the three groups with low motivation, high motivation, and with the control group (see Table 5). In the highly motivated group, (*X*^2^(2, *N* = 127) = 5.59, *p* = 0.06) more participants had a decrease in chronic stress, in contrast to participants in the control group and those with low motivation. Furthermore, significantly *(X*^2^(2, *N* = 127) = 12.26, *p* = 0.01) more participants of the highly motivated group reported a decrease in vital exhaustion when compared to participants in the control group and to participants with low motivation. Finally, there was a decrease in symptoms of depression in a significantly *X*^2^(1, *N* = 127) = 14.44, *p* = 0.01) higher proportion of participants with high motivation compared with participants in the control group and with low motivation.

## 4. Discussion

This study aimed to investigate the effectiveness of resilience training on the containment of chronic stress, symptoms of depression and vital exhaustion, and to evaluate the influence of motivation on training effectiveness. The results showed that resilience training had a favorable impact on mental health outcomes, but almost only under the condition that participants were motivated for the training.

The present study expands upon the current literature. First, results showed some direct training effects with small effect sizes; second, training motivation showed a significant effect on training outcomes.

We formulated two hypotheses and each of them is considered now in turn.

The first hypothesis predicted that resilience training participation will decrease chronic stress, symptoms of depression and vital exhaustion compared to the control group. Data did confirm this for symptoms of depression. Participants of the IG showed a decrease in symptoms of depression after the training, whereas participants in the CG did not. But there were no significant differences between IG and CG for chronic stress and vital exhaustion. Therefore, the present results align with other studies that found only minor effects of resilience training [19], or even non-effects [39], on symptoms such as mental disorders, alcohol use, homesickness and mental health stigmatization. Hence, it was conceivable that further latent but unassessed factors affected the effectiveness of resilience training programs.

The second hypothesis predicted a positive influence of motivation for resilience training outcomes, and the data confirm this. Motivation had a significant effect on all training outcomes. Participants with high motivation for training were able to benefit the most from the resilience training. With the acquired training tools, participants could decrease the vital exhaustion and symptoms of depression significantly. In addition, the training acted as a stress buffer and helped the participants stabilize the chronic stress. In contrast, low motivated participants reported an increase in chronic stress and had no changes in vital exhaustion and symptoms of depression. A larger proportion of highly motivated participants decreased vital exhaustion and symptoms of depression as an indicator of clinical significance than participants with low motivation and than participants in the control group. The relevance of these results was that participants with low motivation showed the same course as participants in the control group, in contrast to participants with high motivation. These results revealed that motivation appeared to be an essential prerequisite for training outcomes; accordingly, no training effect was observed among those participants not motivated for training. Or, in other words, if participants were not motivated for this kind of training, it appeared that it made no sense to register them. Given this, the present findings helped to explain the results of studies that found no or only weak effects of resilience training [19,37,39]. Such effects could be biased by motivation, especially in organizations where the participants did not voluntarily participate in the training.

The medium effect sizes considering motivation showed the great potential of such training, especially regarding the weak effects found in other resilience training studies [19]. Although the comparability between studies is difficult, the present study indicated that the content of the resilience training was optimally tailored to participants of the present sample: Participants benefitted maximally from the training because of a perfect needs assessment performed beforehand and because the training was conducted based on individual examples.

Strikingly, participants in the intervention group had much higher initial scores in chronic stress, vital exhaustion and depressive symptoms when compared to participants in the control group. Since the military training set-up did not allow for a neat separation of these two groups in the same military class, the IG and the CG were conducted at different times. This temporal shift could have led to biased data through external conditions like different group leaders, different military accommodations, or weather conditions. The weather, in particular, could be primarily responsible for different pretest scores because the IG entered officers’ school in winter, while the CG started in the spring. As most military exercises take place outdoors, and given that the winter in Switzerland might be very cold and wet, the data could be biased by weather conditions [68,69]. The difference values in chronic stress, vital exhaustion and depressive symptoms at pretest between lowly and highly motivated participants could be due to the fact that highly motivated participants were not only highly motivated for resilience training (which was a part of OS), but also for the entire OS, which could have resulted in more efforts during the first challenging week of OS. Despite these significant differences in the pretest, there were no more differences in any post-test variable. This finding was another indicator of the effectiveness of the training.

In regard to attrition from military services, the attrition rate of 15% was consistent with previous resilience training research among high-stress cohorts like military personnel [27,70] and police officers [25]. Furthermore, pretest scores between those who continued the OS and those who dropped out demand particular attention: Cadets who dropped out had higher scores in chronic stress and symptoms of depression at pretest. Therefore, it is possible that participants who already had higher stress, symptoms of depression, and vital exhaustion scores at the beginning of the officers’ school had less resources to call on to cope with the stressful military training requirements and thus dropped out.

In our opinion, the present results are of practical importance. The results indicated that motivated participants of the resilience training demonstrated a better resilience in terms of adaptation to stressful circumstances than participants in the CG. The intervention was beneficial and contained essential information on dealing with stress. The imparted tools of resilience training helped to stabilize chronic stress and to reduce vital exhaustion and symptoms of depression in this study. Additionally, the learned tools functioned as a buffer against increasing chronic stress, and it is possible that the effects of training are activated over time. Furthermore, implementing a resilience training during a stressful time like officers’ school appeared useful. This setting allowed participants to practice learned skills directly during exposure to a stressor which results in good training effects.

The novelty of the results should be balanced against the following limitations: First, participants were screened and healthy young men. in this view, when compared with older adults, may be more open to psychological training content [71]. This effect was shown once: Compared with participants at the age of 25 years and older, 18–24 years old participants showed a substantial treatment effect on emotional and social fitness [72]. Second, the lack of strict randomization may have led to a bias between groups [73]. Third, we relied on self-reports, while experts’ ratings might have yielded further important information. Self-reports bear the risk of social desirability [74] or careless responding [75]. Fourth, the study did not investigate the long-term sustainability of the observed effect. Future research should examine the long-term effect of resilience training on chronic stress and stress-related problems, especially because a meta-analysis pointed out that the effects of resilience training weaken over time [19]. Therefore, it is of interest whether the effects among highly motivated persons last longer or are more robust than for low motivated participants.

## 5. Conclusions

This study suggests that in a military context resilience training could reduce stress-related health problems like symptoms of depression and vital exhaustion and buffer chronic stress. However, the effectiveness is only present when participants were motivated for the training. Therefore, all organizations that conduct resilience training with their members should consider participants’ level of motivation. Sufficient energy and time should be invested in conveying meaning and motivation, since training motivation is a crucial component of the training’s success.

## Figures and Tables

**Figure 1 ijerph-19-06179-f001:**
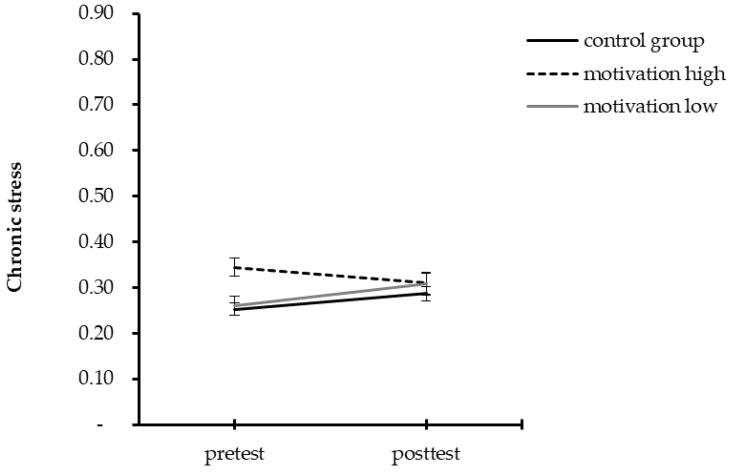
Mean summary for pre- and post-test scores for chronic stress separated by groups.

**Figure 2 ijerph-19-06179-f002:**
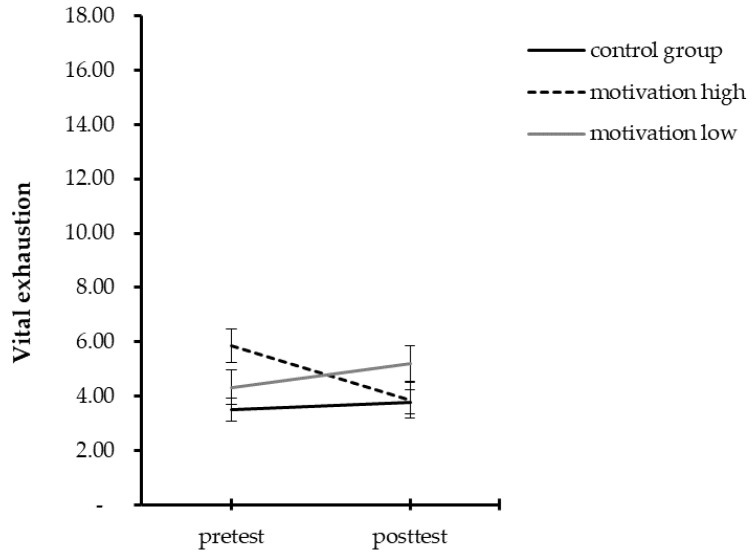
Mean summary for pre- and post-test scores for vital exhaustion separated by groups.

**Figure 3 ijerph-19-06179-f003:**
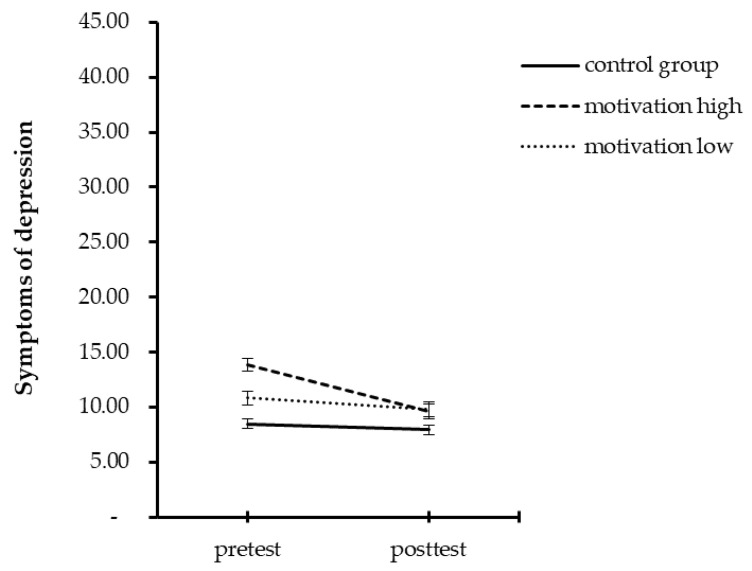
Mean summary for pre- and post-test scores for symptoms of depression separated by groups.

**Table 1 ijerph-19-06179-t001:** Sample characteristic.

	Groups		Statistics
	Intervention	Control	
	*M (SD)*	*M (SD)*	
N	62	65	
Age (in years)	20.85 (1.32)	21.03 (1.80)	*t*(117.40) = −0.63, *p* = 0.529; *d* = 0.11)
Education Level	*n* (%)	*n* (%)	*χ*^2^ (N = 127; *df* = 1) = 0.162, *p* = 0.69
Upper secondary school	96.8%	95.4%	
Tertiary level	3.2%	4.6%	

**Table 2 ijerph-19-06179-t002:** Descriptive and inferential statistical indices for chronic stress, vital exhaustion and symptoms of depression, at pre- and posttest, separately for the intervention and control groups.

	Time Points				Factors					
	Pretest		Posttest		Time		Group		Time × Group Interaction	
	IG	CG	IG	CG						
	*M (SD)*	*M (SD)*	*M (SD)*	*M (SD)*	*F*	*η_p_* ^2^	*F*	*η_p_* ^2^	*F*	*η_p_* ^2^
Chronic stress	0.31 (0.13)	0.25 (0.11)	0.31 (0.13)	0.29 (0.13)	3.71	0.029 ^[^^S^^]^	3.60	0.028 ^[^^S^^]^	3.17	0.025 ^[^^S^^]^
Vital exhaustion	5.11 (3.63)	3.51 (3.39)	4.50 (3.41)	3.78 (3.89)	0.41	0.003 ^[^^T^^]^	4.01 *	0.031 ^[^^S^^]^	2.84	0.022 ^[^^S^^]^
Symptoms of depression	12.40 (7.23)	8.48 (4.44)	9.68 (6.73)	7.92 (5.43)	9.68 **	0.072 ^[^^M^^]^	9.26 **	0.069 ^[^^M^^]^	4.25 *	0.033 ^[^^S^^]^

*Note*: *N*_IG_ = 62, *N*_CG_ = 65; * *p* < 0.05, ** *p* < 0.01; ^[T]^ = trivial effect size, ^[S]^ = small effect size, ^[M]^ = medium effect size; IG = intervention group, CG = control group.

**Table 3 ijerph-19-06179-t003:** Correlations among study variables.

Variable	1 ^a^	2 ^b^	3 ^b^	4 ^b^	5 ^b^	6 ^b^	7 ^b^
1. Motivation							
2. Chronic stress pre	0.20						
3. Chronic stress post	−0.05	0.66 ***					
4. Vital exhaustion pre	0.14	0.53 ***	0.49 ***				
5. Vital exhaustion post	−0.17	0.46 ***	0.62 ***	0.66 ***			
6. Symptoms of depression pre	0.10	0.62 ***	0.55 ***	0.44 ***	0.40 ***		
7. Symptoms of depression post	−0.14	0.66 ***	0.65 ***	0.34 ***	0.51 ***	0.53 ***	

*Note*. ^a^ *N* = 62; ^b^ *N* = 127; *p* < 0.01, *** *p* < 0.001.

**Table 4 ijerph-19-06179-t004:** Descriptive and inferential statistical indices for chronic stress, vital exhaustion and symptoms of depression, at pre- and posttest, separately for motivation high (M_high_), motivation low (M_low_), and for the control group (CG).

	Time Points						Factors					
	Pretest			Post-test			Time		Group		Time × Group Interaction	
	M_high_	M_low_	CG	M_high_	M_low_	CG						
	*M (SD)*	*M (SD)*	*M (SD)*	*M (SD)*	*M (SD)*	*M (SD)*	*F*	*η_p_* ^2^	*F*	*η_p_* ^2^	*F*	*η_p_* ^2^
Chronic stress	0.34 (0.12)	0.26 (0.12)	0.25 (0.11)	0.31 (0.14)	0.31 (0.12)	0.29 (0.13)	1.91	0.015 ^[^^T^^]^	2.76	0.043 ^[^^S^^]^	6.17 **	0.091 ^[^^M^^]^
Vital exhaustion	5.84 (3.91)	4.33 (3.19)	3.51 (3.39)	3.84 (3.14)	5.20 (3.60)	3.78 (3.89)	1.16	0.009 ^[^^T^^]^	1.99	0.031 ^[^^S^^]^	9.66 ***	0.135 ^[^^M^^]^
Symptoms of depression	13.87 (8.42)	10.83 (5.39)	8.48 (4.44)	9.59 (7.90)	9.76 (5.35)	7.92 (5.43)	12.72 ***	0.093 ^[^^M^^]^	5.21 **	0.078 ^[^^M^^]^	4.52 *	0.068 ^[^^M^^]^

*Notes*: *N*_Mhigh_ = 32, *N*_Mlow_ = 30, *N*_CG_ = 65; * *p* < 0.05, ** *p* < 0.01, *** *p* < 0.001; ^[T]^ = trivial effect size, ^[S]^ = small effect size, ^[M]^ = medium effect size.

**Table 5 ijerph-19-06179-t005:** Reliable change index.

Outcome		M_high_ (*n* = 32)	M_low_ (*n* = 30)	CG (*n* = 65)	*Fisher’s Exact Test,* Cramér’s *V*
Chronic Stress	Increase	3%	7%	8%	*p* = 0.799, *V* = 0.078 ^[S]^
	Decrease	9%	0%	2%	*p* = 0.111, *V* = 0.210 ^[S]^
Vital exhaustion	Increase	0%	3%	5%	*p* = 0.672, *V* = 0.109 ^[S]^
	Decrease	13%	0%	0%	*p* = 0.006, *V* = 0.311 ^[M]^
Symptoms of depression	Increase	3%	3%	2%	*p* = 0.800, *V* = 0.056 ^[S]^
	Decrease	25%	3%	3%	*p* = 0.002, *V* = 0.337 ^[M]^

*Notes*. M_high_ = motivation high, M_low_ = motivation low, CG = control group ^[S]^ = small effect size, ^[M]^ = medium effect size.

## Data Availability

Data belong to the Swiss Armed Forces; data are made available to experts in the field upon request and upon the detailed description of the reason of request.

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
