# Peer review of "Does Training Motivation Influence Resilience Training Outcome on Chronic Stress? Results from an Interventional Study"

_ijerph, 2022, doi:10.3390/ijerph19106179_

Round 1
Reviewer 1 Report
Manuscript ID ijerph-1702347
Type Article
Title
Does Training Motivation Influence Resilience Training Outcome on Chronic Stress? Results From an Interventional Study
General comments
- Nicely written ms without spelling errors and almost no typos. The specific points listed in the table below in this review are mostly idiomatic rather than substantive and should be fairly readily sorted out.
- Good description and presentation of results. Discussion avoided over-assertion which was good. I appreciated the candid pointing out of limits to generalising the findings – well done.
- I wondered in the exclusions, given the need to exclude significantly unmotivated or unserious potential participants, whether an additional short comment might be made in the discussion about whether this might have constrained the overall significance of the findings. Just an idea – I’m always interested in what the marginal cases say about overall results.
Specific points
|
Line 21 |
Typo – small t and capital T present for ‘tThe’ |
|
Line 51 |
I don’t ‘get’ the use of ‘plausibly’ in this line? |
|
Line 54 |
Suggest replacing ‘on dealing with’ with ‘about’ |
|
Line 56 |
‘Therefore” in this line is making too strong a causal connection from the previous sentence – softening your claim will make it more acceptable. Even better, just drop the word ‘therefore’ and make it a second separate statement, like you do in the next sentence. |
|
Line 67 |
‘Next’ doesn’t seem to add any value in this line. |
|
Line 71 |
Again, ‘Therefore’ is not needed. |
|
Lines 88-89 |
Again too strongly asserted. Need to revise to say something like - has elements that support a more objective assessment… [1/ nothing is ever fully objective, and 2/ the army as a specific setting can have strong counter-objective hierarchies and cross-pressures] |
|
Line 92 |
Should ‘and’ be ‘but’ as more appropriate? |
|
Line 197 |
‘neither’ is not used idiomatically here. |
|
Line 115 |
‘did investigate’ would be more idiomatic as ‘has investigated’ or ‘had investigated’ |
|
Line 118 |
Suggest replacing ‘of’ with ‘to’ |
|
Line 119 |
Replace second use of ‘dysfunctional behavior’ with ‘it’ |
|
Line 222 |
Suggest replacing the comma and ‘if’ with ‘whether’ |
|
Lines 277-279 |
Insert ‘the’ before ‘intervention’ and again before ‘control’ |
|
Lines 270, 281, 290, 316, 337 |
Depending on editor’s wish, I would suggest left-justifying the first column of the tables |
|
Line 276 |
Place hyphen again the ‘pre’ as you do in the next line. |
|
Line 346 |
I do not know what the phrase ‘To the following two reasons” means – suggest deleting. Also suggest changing ‘expand’ to expands’ |
Author Response
We thank Reviewer # 1 for their care devoted to thoroughly review the present manuscript. The comments and suggestions helped us a lot to improve the quality of the manuscript. Please find the detailed point-by-point-response attached as a separate file. Again, thank you so much for all your kind efforts.

Reviewer 2 Report
The originality of the study is related to the specificity of the study group. The study is interesting and important due to the social role that students (professional military, officers) are prepared for.
For the quality of service and effective performance of tasks, it is necessary for soldiers to be resilient mentally, to be able to cope with stress and to be mentally healthy. From this point of view, the research has great practical value. In general, the results of other studies on stress, chronic stress resistance training and their association with depression, exhaustion and motivation to participate in training have been confirmed. It should be emphasized, inter alia, sequence result: strong motivation - stabilization of chronic stress levels over time - decrease in the severity of symptoms of depression and exhaustion. It would be interesting to refer to the "old" findings of Yerkes-Dodson on the strength of stress and the effectiveness of coping with tasks of various difficulties. If these findings are still accurate, it would mean that the stress of future soldiers with a strong motivation for training (resistance training) was low - because the tasks set for them were not difficult for them. Undoubtedly, it is worth researching other variables - as mentioned by the authors - maybe the intellectual level? Or maybe the difficulty of training that soldiers go through, and probably the interaction of both factors.
I am wondering about the description of the Y axis in Figures 1, 2 and 3. Is it about the values ​​/ levels of variables (e.g. the severity of depression) - or the magnitude of the changes in the severity?
Another question that arises is whether low or unshakable motivation to train - although it is associated negatively with the strength of chronic stress (intensification over time) - is not associated with depression and exhaustion - going further - does this mean that the change in the strength of chronic stress - is not associated with changes in exhaustion and depression?
So it is possible that being in the "special" (trained) group could be a highly stressful factor?
There is also possible an explanation that in the group of strong motivation - the effects of training are activated over time.
My comments, suggestions - for consideration by the authors (authors' decision)
Author Response
We thank Reviewer # 2 for their care devoted to thoroughly review the present manuscript. The comments and suggestions helped us a lot to improve the quality of the manuscript. Please find the detailed point-by-point-response attached as a separate file. Again, thank you so much for all your kind efforts.

Reviewer 3 Report
The manuscript," Does Training Motivation Influence Resilience Training Out- 2 come on Chronic Stress? Results From an Interventional Study" presents an interesting approach to the collection information of the psychological impact of training programo on cadets in the Swiss Armed Forces.
The “Introduction” section of the manuscript provide extensive revision and with a very good redaction. The review of literature is relevant to their study.
The aim of the study is properly highlighted and justified.
The manuscript, using cuantitative and cualitative techniques based in different cuestionnaires and intervention program. The presentation of the technique and characterization of the results achieved indicate that the method is quite suitable and in fact could be useful to profundize this aspects in the the health sciences.
However, regarding the sample collection and procedure, I would like to receive some additional information:
1. Did the participants voluntarily agree to participate, were they paid?
2. Could you specify any activities that were carried out for the intervention program? any examples of these?
Overall the results are compelling and indicate that the method is more than suitable. Rigorously, the analisys are detailed. The authors do a very good job of presenting a methodology of the accuracy and precision of their results and demonstrate the suitability of the method. Further, the manuscript presents a good and actualized bibliography. The study is of interest for the scientific community.
Author Response
We thank Reviewer # 3 for their care devoted to thoroughly review the present manuscript. The comments and suggestions helped us a lot to improve the quality of the manuscript. Please find the detailed point-by-point-response attached as a separate file. Again, thank you so much for all your kind efforts.

Reviewer 4 Report
This is an interesting study and it is within the remit of International Journal of Environmental Research and Public Health, but it should be revised before it can be accepted for publication.
I have made some recommendations/comments below:
- I suggest to revised the introduction and to use a be more concise in illustrating the literature in order to make it easier for the reader to read.
- I suggest to clearly state in the text that the term stress-related mental health problems refers to depressive symptoms and vital exhaustion or to eliminate the term stress-related mental health problems (that can be confusing) and use the name of two dimensions actually investigated.
- Line 159-161 “Due to the strict 159 Swiss Armed Forces selection process, these cadets consisted of a representative sample 160 of healthy young Swiss men.” Precisely because it is a highly selected sample I would use more caution in this statement.
- The work is based on the application of an intervention to promote resilience in a context considered very stressful, surprisingly the Authors have not included a pre-post intervention measure of resilience. Why hasn't it been evaluated?
- Table 3 I think that the correlation between motivation and the other dimensions investigated should be conducted separately for groups IG and CG respectively, especially as regards post intervention measures.
- Only in the discussions did the authors clearly report that the two groups have significantly different pre-test values ​​between groups in the direction of greater symptomatology in the experimental group. I understand all the possible motivation given by the Authors but this is a very important limit. This information should also be reported immediately in the results as it represents a very serious limitation to the interpretation of all subsequent results. An important element to apply the anova for repeated measures is to consider the two groups homogeneous at the start but if they are several bias can influence the reading of the results. I suggest to test an analysis that takes into account the starting differences between groups.
Author Response
We thank Reviewer # 4 for their care devoted to thoroughly review the present manuscript. The comments and suggestions helped us a lot to improve the quality of the manuscript. Please find the detailed point-by-point-response attached as a separate file. Again, thank you so much for all your kind efforts.

Round 2
Reviewer 4 Report
I have carefuly read the response to reviewers and the revised manuscript, I have found the manuscript improved, the Authors addressed the issues that can be modified and provided clarifications for the ones that can not be modified.
I think the manuscript can be accepeted for publication.